# A Framework for Developing Green Building Rating Tools Based on Pakistan's Local Context

**Muhammad Afrasiab Khan \*, Cynthia Changxin Wang and Chyi Lin Lee** 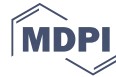

School of Built Environment, University of New South Wales, Kensington, NSW 2052, Australia;
cynthia.wang@unsw.edu.au (C.C.W.); chyilin.lee@unsw.edu.au (C.L.L.)
\* Correspondence: afrasiab.rao@gmail.com or muhammad.khan2@unsw.com.au; Tel.: +61-4133-25-766

**Abstract:** Most countries have developed green building rating tools that are based on social, environmental, and economic dimensions. Pakistan followed a similar approach and has developed a rating tool known as Sustainability in Energy and Environmental Development (SEED). However, SEED is built on developed western countries' rating tool standards which do not address Pakistan's unique local context, especially from the cultural and governmental perspectives. This research aims to fill this research gap by developing a holistic framework of building rating tools that incorporates cultural and governmental dimensions. Based on an extensive literature review, a hypothetical framework, incorporating Pakistan's unique local contexts and adding cultural and governmental dimensions to the widely adopted social, environmental, and economic dimensions of sustainability, was proposed in this paper. This framework was further validated by in-depth interviews with multiple stakeholders in Pakistan. A qualitative analysis of the interview results was carried out, and the final framework was proposed with key indicators, reflecting all five dimensions of sustainability. The verified sustainability framework can be used to improve or develop green building rating tools for Pakistan, and it can also inform other developing countries' rating tool development.

**Keywords:** green building; rating tools; sustainability indicator; SEED; construction industry

## 1. Introduction

The construction industry is vital to the broader economy of a country having several backward and forward connections with socio-economic development, sustainable growth, and environment-friendly infrastructure. According to the World Business Council for Sustainable Development, building can have significant adverse impacts on the environment, consuming 40% of the total energy which generates greenhouse gases (GHG) and appearing to be one of the key contributors to global warming. By 2035, buildings will emit 42.4 billion tons of carbon globally with an increase of 43% since 2007 [1]. Buildings, therefore, can play a critical role in reducing carbon emissions and reducing the adverse impacts of global warming [2–7]. Moreover, buildings and construction materials are extremely imperishable, and therefore, they continue affecting society and the environment for the long term. Over the past decade, the green building concept has been widely recognized, and emphasis has been laid on developing various building rating tools to gauge the green building criteria. Madson and Campbell [8] state that building rating tools provide systematic frameworks to evaluate the performance criteria, thus enabling the buildings to be measured and compared to promote the movement towards more sustainable forms of designing, constructing, operating, and dismantling buildings. These tools help policymakers to make conscious decisions to make society more sustainable [9,10]. Building rating tools are one of the most complex types of appraisal methodologies as it entails multidisciplinary aspects, i.e., environmental, economic, social, cultural, and value-based elements [11,12]. According to the British Research Establishment (2008), there are more than 600 rating tools reported worldwide [13,14]. For instance, BREEAM-UK, LEED-USA, GBI-Malaysia, GREEN STAR-Australia, and GREEN GLOBES are famous certification standards used

around the globe. Regarding this, Pakistan has also developed its own GB rating tool, known as SEED, (Sustainability in Energy and Environmental Development) [15]. Hence, building rating tools specific to a country are important to address the local environmental needs and sustainability issues.

Different studies highlight that there is a lack of incorporation of local conditions in building assessment tools. Lin and Ling [16] and Ajayi et al. [17] stated that most of the existing rating tools were established in developed countries with cold climates, stable economies, and different social values. It is not practical to implement their latest sustainable technologies in a developing country like Pakistan with different climatic conditions. Moreover, there is a lack of a holistic approach and comprehensive coverage of the sustainability issues. Research studies of building rating tools illustrate that the existing rating tools mainly cover the environmental aspect with little consideration to social and economic impacts [18,19]. Furthermore, there is an excessive number of tools and methods available to rate and evaluate the sustainability of buildings globally with no international guidelines. In addition to this, the assessment methods utilized by green building rating tools and life cycle assessment tools are not incorporated into each other. Therefore, the green and sustainability rating systems are a greenwashing activity and are often used for impression management in which the systems have become a sustainability mask, misleading its targets and initial intended purposes [19,20].

In Pakistan, no comprehensive assessment tool has been initiated by the government or local authorities to evaluate sustainable development. The developed rating tool SEED is based on western standards and is not able to address the local environmental needs and sustainability concerns [21]. Pakistan has different climatic factors, weather parameters, and variables that need to be analyzed and addressed in the green building rating tool, which has not been done previously. So, the certification standard needs to be revised to be specific to Pakistan. Furthermore, there are only a few buildings that are green certified in Pakistan and this is not implemented at large. There is a need to develop policy and legitimate guidelines for the promotion and implementation of sustainable development assessments. Therefore, this current study aims to address these challenges and develop a comprehensive green building framework for Pakistan. This is the first study to develop a holistic framework of building rating tools that consider the uniqueness of Pakistan's local context. This offers an enhanced understanding of building rating tools for policymakers to develop green rating tools and facilitate the implementation of sustainable developments in Pakistan and other developing countries. More specifically, a hypothetical framework is developed based on an extensive literature review. This framework is validated through in-depth interviews with multiple stakeholders in Pakistan. A qualitative analysis of the interview results will be carried out and the final framework will be proposed with key indicators and significant dimensions of sustainability. After validation, a comprehensive framework deemed suitable for the local context of Pakistan is provided for the building rating tool of Pakistan.

This study contributes to the literature by developing a building rating tool framework with an emphasis on the very humanistic factor of "Life Cycle Thinking" for the first time. It is important because it refers to the very human dimension of existence and habitation, which, as this study finds, has a huge and obvious impact on the quality of the assessed objects and urban structures. In addition, this study also compares criteria used in the preparation of building ratings in Pakistan (SEED) and internationally (e.g., GREENSTAR, LEED, BREEAM, CASBEE, DBNG). Using the example of regulations adopted in Pakistan, this study attempts to take a critical look at this documentation, making use of various expert opinions. Importantly, this study builds an even more thorough inspection system, including all social and cultural aspects, which have been marginalized in existing certification systems. The findings of this study are critical to policy makers. Importantly, these contribute to the development direction of green buildings as green building rating tools evolve over time [22,23] and reduce green washing activities (Guo, Zha et al. and Khan, Sepasgozar et al. [20,24]).

## 2. Materials and Methods

The methodology adopted for this study comprised of (1) literature review and analysis, for the development of the conceptual framework, and (2) qualitative data collection and analysis for the validation of the conceptual framework. The methodology for establishing assessment criteria for sustainable buildings and construction aligns with earlier studies [25–31]. An extensive review of the literature was carried out for this study (Table 1). The literature review provides a complete understanding of the related literature in green building rating tools and identifies the gap in the literature. It retrieved maximum number of relevant articles based on this study. After retrieving the articles by keyword search through different search engines a screening was carried out to further narrow down the selection criteria. The articles were screened based on (1) no duplicates; (2) publication period 2010–20; (3) document type was restricted to research article and book chapter; (4) published in English language only. Based on these metrics, the relevant, applicable, and unique research articles were extracted. Based on scientific and theoretical knowledge from the literature, a theoretical framework was developed focused on different dimensions of sustainability. Several indicators were identified and grouped under each dimension in the theoretical framework and were related to the assessment stages of the building life cycle. To validate this framework, feedback from the respondents was collected through semi-structured interviews for further refinement of indicators.

**Table 1.** Articles retrieved and shortlisted for final content analysis.

| Categories | Articles Retrieval | Filtered Articles | Final Content Analysis |
|---|---|---|---|
| Green Building Rating Tools | 1120 | 122 | 85 |
| Sustainability Assessment of Buildings | 1225 | 325 | 149 |
| Total | 2345 | 447 | 234 |

Based on a refined list of indicators, a questionnaire was designed to conduct semi-structured interviews. Semi-structured interviews could be used to seek more information through informal and spontaneous probes when possible, to gain more valuable information [1]. The interview questionnaire was composed of a mix of both structured and unstructured questions. The unstructured questions were open-ended so that respondents were given more space and freedom to share their perceptions, which was an effective way to gauge the perception and opinion of humans [32–34]. The semi-structured approach offered flexibility to the participants and provided consistency in the interviews to avoid going off the topic. This approach also made it possible for the researcher to verify the response thoroughly and gather if any additional information was required [35,36]. Semi-structured interviews were conducted with 25 building stakeholders, for validating the developed theoretical framework. A sample size of 20 to 30 is deemed adequate to enable internal generalization in a qualitative study [37]. To increase the sample diversity and avoid biases, a set of three groups of participants were chosen, including (1) rating tool developers, (2) experienced professionals with 10+ years of industry experience, and (3) experienced academicians with 15+ years of academic experience. The first section of interview questions was related to the background information of the participants, to ensure that the selected participants fulfil the selection criteria (supplementary information Table S1). After the background section, all the questions that were asked of the participants were based on local context, building life cycle thinking, dimensions of sustainability, and framework indicators. These questions are explained in the result section with their key objective.

In the last step, the recorded interviews were transcribed into the text data. The transcribed data from the interviews was analyzed using content analysis which is a process of "identifying, coding, categorizing, classifying, and labelling the primary patterns in the data" adopting the method of [33]. This analysis process was done using NVivo version 11 which enabled categorization of information into relevant themes.

## 2.1. Literature Review

Sustainable assessment is based on the conceptual framework which provides a basis for identifying key criteria and indicators [11,33]. The historical evolution of the perception and measurement of sustainable development has evolved over the years. Some of the key frameworks include the United Nations Commission on sustainable development in 1992 based on 68 indicators that covered social, environmental, economic, and functional features of sustainable development. In 1996 a Consultative Group on Sustainable Development Indicators (CGSDI) was established that focused on establishing a single sustainability index. This resulted in developing a "dashboard of sustainability" for over 100 countries having 46 indicators structured into four clusters, i.e., environment, economy, society, and institutions. Similarly, the development of wellbeing assessment for 180 countries was sponsored by World Conservation Union (IUCN). The index is composed of 88 indicators that have subindexes of human wellbeing and ecosystem wellbeing. The World Economic Forum's Environmental Sustainability Index is developed for 148 countries based on 68 indicators. While Global Scenario Group is composed of 65 indicators including aspects of national and international equity, deforestation, toxic waste, carbon emissions, etc. [6,7].

To develop a systematic framework for sustainability analysis of infrastructure, two benchmarks are marked, i.e., "work" and "nature" [38]. "Work" benchmark means the behavioral relationships among the construction products and to study technical, social, cultural, and economic sustainability. Moreover, the "nature" benchmark defines the interactions between construction processes and the surrounding ecosystems to study environmental, individual, and economic sustainability [39]. Current assessment frameworks are categorized based on these two basic benchmarks and are divided into two basic concepts [40,41]. Some assessment frameworks are based on the triple bottom line concept by utilizing the qualitative method to address sustainable development. Many international tools have been developed based on the triple bottom line concept using the qualitative method, i.e., LEED from the US, BREEAM from the UK, GREEN STAR from Australia, and CASBEE from Japan, also called criteria-based tools, while some of the assessment frameworks are based on life cycle thinking, utilizing the quantitative method to address sustainable development. Numerous researchers have considered the ways of addressing these gaps in the existing assessment frameworks by adding new social and economic indicators to make them less environmentally focused. These concepts are discussed in detail in the following sections.

### 2.1.1. Dimensions of Sustainability

According to Salahuddin and Gow [42], buildings which address the environmental issues can only be stated as green buildings and are sustainable if they incorporate the social and environmental issues (Figure 1). Hence, an assessment process incorporating all three dimensions of the triple bottom line model addresses sustainability more holistically. Hence, the rating tools that initially assessed environmental dimension only are shifting to incorporate more social and economic indicators to be assessed to provide a more holistic assessment of the performance of the buildings [24].

The triple bottom line (TBL) concept has been criticized as a limited tool for understanding sustainable development and covering all underlying issues [43]. To address this limitation, researchers have incorporated more dimensions such as governance and culture [44–47]. Table 2 provides a list of studies that challenge the effectiveness of the triple bottom line and have incorporated one or more dimensions to the existing triple bottom line according to the relevant fields of study.

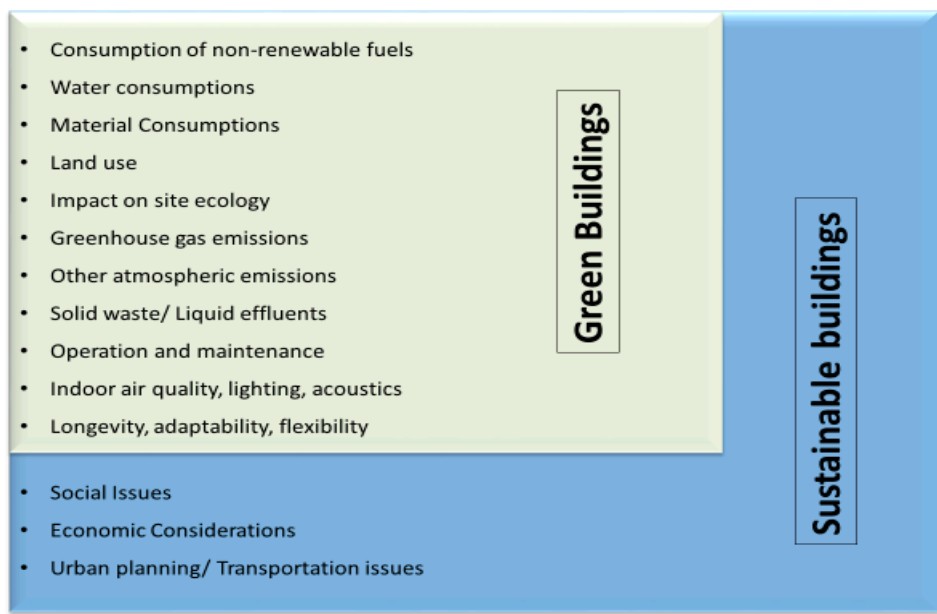

**Figure 1.** Building performance goal defined by the scope of issues considered (UNEP 2009).

**Table 2.** List of literature extending the triple bottom line.

| Dimensions of Sustainability | Author |
| --- | --- |
| Social, Economic, Environment, and Culture | Scrimgeour [48] |
| Social, Economic, Environment, and Spirituality | Daniel [49] |
| Social, Economic, Environmental, and Governance | Woermann and Engelbrecht [50] |
| Social, Economic, Environmental, and Sensitivity | BRATT [51] |
| Social, Economic, Environmental, and Cultural | Soini, Jurgilevich [52] |
| Social, Economic, Environmental, and "Personal and Family Happiness" | Fonseca [53] |
| Social, Economic, Environmental, and Cultural | Walters, Johnson-Jennings [54] |
| Social, Economic, Environmental, Cultural, and Political | Thornton [55] |
| Social, Economic, Environmental, and Relevance | Bodner [56] |
| Social, Economic, Environmental, and Perception Politics | O'Neil [57] |
| Social, Economic, Environmental, and Governance | Alibašić [58] |
| LCSA = LCA + LCC + SLCA | Mahbub, Oyedun [59] |
| Social, Economic, Environment, and Governance | Mokoena [60] |
| Social, Economic, Environment, and Governance | Salahuddin and Gow [42] |
| Social, Economic, Environment, Culture, and Governance | Karaca, Guney [61] |
| LCSA = LCA + LCC + SLCA + Culture LCA | Pizzirani, McLaren [62] |

### 2.1.2. Governance Dimension

The quality of governance plays a defining role in supporting the economic, social, and environmental dimensions of the sustainable development goals [63]. The role of governance has gone under-emphasized in the triple bottom line, but it is very significant to lead the way towards sustainable development [57,58]. Goss [64] defines governance as "emerging forms of collective decision-making at a local level, leading to the development of different relationships, not only among public agencies but also with citizens". Kemp [65] expressed that governance practices can foster and guide progress towards sustainability. Alibašić [58] argues that Elkington's triple bottom line was established for the private sector; however, its application to other sectors is possible by broadening the base with an added perspective of "governance". Alibašić [58] introduced the four-pillared Quadruple bottom line while stressing that "governance is a dynamic component necessary to the success of the sustainability". Simons [66] also indicated that the quadruple bottom line approach is suitable for the urban development sustainability assessment.

### 2.1.3. Culture

While some studies consider governance as an essential component for sustainability, in many other studies, culture is measured as an alternative fourth dimension in addition to the triple bottom line. Culture is defined as a dynamic mix of symbols, beliefs, languages, and practices that people create, not a fixed thing or entity governing humans [67]. Cultural sustainability was introduced by Hawkes [68] as a dimension of sustainability with a focus on local planning. United Cities and Local Governments (UCLG) has promoted culture as a dimension of sustainable development with inclusion in the local policy process. UCLG adopted Agenda 21 for Culture, by importing culture into the Rio + 20 process [5]. Numerous studies stressed the significance of culture and human factor in sustainable development [69–71], whereas cultural sustainability has been examined from a community development perspective [72]. Besides, Camarda [73] explored the post-industrial regeneration of communities regarding cultural sustainability. Despite the continuous acknowledgement of culture as a dimension in sustainable development [74,75], it has been rarely incorporated [62]. International, national, regional, and local policy aimed at sustainable development often examines the cultural dimension as part of the social one or completely ignores it [76,77]. Pizzirani, McLaren [62] also differentiated the social indicators from the cultural once as shown in Table 3. Isar [77] has also emphasized the acceptance of heritage and culture as the fourth dimension of sustainability parallel to social, ecological, and economic sustainability. Several other researchers have stressed on consideration of the cultural dimension in the triple bottom line and move it to the quadruple bottom line [56,78–80].

**Table 3.** Differentiation of social and cultural indicators [62].

| Social Indicators | Cultural Indicators |
| --- | --- |
| 1. Health | |
| 2. Political participation | 1. Language |
| 3. Safety | 2. Autonomy |
| 4. Housing | 3. Media communication |
| 5. Skills | 4. Artistic creation |
| 6. Social connectedness | 5. Protection of culture and places |
| 7. Recreation and tourism | |

### 2.1.4. Quintuple Bottom Line Approach

As evident from the literature, both governance and cultural dimensions have a significance about sustainability; especially when considering the sustainability issue in a developing country such as Pakistan had a unique culture. Hence, culture and governance should be integrated into the exiting triple bottom line for sustainability study focused on Pakistan and this study proposes for the first time the extension to a quintuple bottom line model. It has a similar ring to its former as shown in Figure 2, with the added acknowledgement of governance and the culture. In an earlier study, a detailed comparison between the achievements and targets for both US and Pakistan with the 17 United Nations sustainable development goals was carried out. It was found that the targets for both countries are different from each other. Pakistan faces major challenges in terms of social and cultural constraints, and the lack of government support to achieve sustainable development is prevalent. The existing rating tools need further research to address the regional context and allocate credit points accordingly [24].

Hence, the importance of such integrations of governance and culture in the triple bottom line is enormous. It carries the significant potential to address the overall sustainability of the built environment and achieving the goals of creating a sustainable future for Pakistan. To create a sustainable building future, this integration, however, will provide limited results unless accompanied by other meaningful concepts like life cycle thinking

and form of innovative sustainability assessment mechanism that can determine the extent to which the development will be considered sustainable.

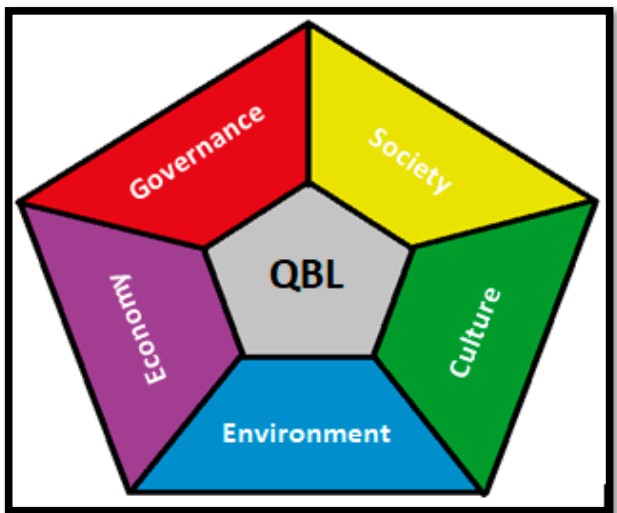

**Figure 2.** Quintuple bottom line approach developed based on an extensive review.

### 2.1.5. 'Life Cycle Thinking' Theory

The lifecycle refers to five main phases of building, starting from the design phase until the construction phase, followed by operation, maintenance phase, renovation phase, and lastly demolition [81–84]. Life cycle thinking emphasizes addressing the individual stages in such a way that results of one stage are being shifted rationally to the other stages. It also allows understanding how each downstream phase is affected by the choices that are made for implementation and development of the previous phases, and how that can be addressed and improved if needed to enhance the sustainability of the building [85]. However, in rating tools after demolition, reuse and recycling phases also need to be considered [86,87]. Each phase is generally managed independently but at the same time interlinked with the rest of the building life cycle. Buildings consume resources, energy, and financial assets throughout their whole lifecycle; thus, they have significant impacts not only on the environment but also on the economy and society [88,89]. Life cycle approaches and assessment methodology are inherently rooted in sustainability valuation at the conceptual level [11]. Although the life cycle approach has been considered in sustainability assessment since the 1990s, the recent developments in life cycle assessment led to a very comprehensive proposal of frameworks for a lifecycle-based sustainability assessment [32]. According to Kloepffer [90], Life Cycle Sustainability Assessment equates to life cycle assessment, life cycle costing, and social life cycle assessment. Hence, if all the stages of the building life cycle are included in the building, the sustainability assessment process will be enriched [34,91].

There has been no single framework, methodology, or approach which has received the consensus on the issue of the sustainability assessment, and the process of achieving a more holistic framework addressing the sustainability issue is still ongoing. Although shifting towards a sustainable world is unavoidable, the path to be followed by Pakistan is different from developed countries. If measures are not taken, it may lead to a state where the result of the analysis may not illustrate the actual scenario of the country or the region. Therefore, a conceptual sustainability assessment framework for Pakistan is required.

### 2.1.6. Key Indicators

The indicator is a measurable parameter, which provides information on, or maps out the state of a phenomenon, with a meaning reaching out beyond that directly associated with its value [92]. The parameter could be quantitative, semi-quantitative, or qualitative and is controlled through a tool.

Some indicators have been proposed by literature, but they have not yet been part of the building rating tool. Economic aspects are discussed by much research [93,94]. Further, Huo [94] discussed broader economic aspects including 'land cost', 'professional fees', 'construction cost', 'operating cost', 'occupancy cost', 'demolition cost', 'salvage value', and 'other costs and charges' which have not been given much consideration. Similarly, Zhang [95], compared BREEAM, LEED, and Chinese standards for green building evaluation and concluded that it is necessary to pay attention to the relationship between the buildings' economic performance and environmental performance, and the evaluation indicators should be in harmony with each other [70]. This shows the need to consider the economic aspects of building development and the clear lack of consideration of economic indicators in credit criteria of almost all the green building rating tools 'resilience (adaptation and mitigation)' criteria related to governance is identified by the study in Malaysia [42] which has been addressed only in CASBEE rating tools.

The tools reviewed for this study cover the life-cycle phases variously, and the coverage of each dimension of sustainability is different and non-comprehensive too. Also, these tools use the same criteria for several dimensions but different indicators. In this case, the questions of which criteria are the most important, which indicators correspond to particular sustainability dimension best, and which indicator addresses specific phase of building life cycle arise [96]. Thus, there is the need to identify the most appropriate sustainability indicators which can address all sustainable issues and incorporate building life cycle addressing different local situations [97]. This would allow the selection of the indicators, relevant to the dimension of sustainability and, at the same time, suitable to the context of Pakistan, to assess sustainability performance at a "local" level.

The guidelines defined in the literature were considered while assigning the indicators into the relevant dimension of sustainability. When indicators represented one or more dimensions of sustainability, the most suitable dimension was selected after considering the intent of this indicator as given in the relevant green building rating tool. This is also in line with the guideline of conducting a good interview as discussed by Gillham [35] and Crowe and Sheppard [36]. For instance, the division proposed in ISO/AWI 21929, energy, atmosphere, materials, resources, and pollution relate directly to the environmental dimension of sustainability [98]. Water efficiency is included in both environmental and economic dimensions. Sustainable sites belonging to indoor air quality comes under the socio-economic aspect. While location and transport belong to all the three dimensions of sustainability [99]. BREEAM, DGNB, and Envest 2 are the rating tools that are considering economic aspects. Some indicators follow a procedure that falls in none of environmental, social, economic, cultural, or governance dimensions but still is assessed. For example, some points can be achieved if a project demonstrates that accredited professional people participate in the project for Green Star, or the integrated design process is recognized for BREEAM and LEED, while CASBEE rewards levels for enough maintenance plans. Approximately 15% of total indicators of Green Star NZ are divided into procedural compared to BREEAM, LEED, and CASBEE with a relatively low percentage. In other words, sub-categories of BREEAM, LEED, and CASBEE are more sufficiently addressing the green or sustainable assessment in which a greater percentage of total subcategories is used to assess sustainable dimensions.

The life cycle approach confirms that improving one type of impact at one stage can have negative impacts on another. This means that the more the impacts of each life cycle are included in the assessment, the more accurate the assessment will be, hence enriching the building's sustainability. The prime resources that are considered in this section are land, water, energy, air, and green cover. The building operation stage involves the issues of operation and maintenance of building systems, monitoring, and recording of energy consumption, occupant health and well-being, and issues that affect the global and local environment. Most of the building rating tools generally cover construction, operation, and maintenance phases; they do not comprehensively include performance indicators referring to each stage of the life-cycle process [17]. Table 4 below demonstrates the division

of indicators in all the five dimensions within eight building rating tools. While based on the analysis of the above literature, a new conceptual framework is developed (Table 5). The purpose of developing this framework is to encourage sustainable building design, construction, operation, renovation, and demolition through better integration of the sustainability dimensions. Hence, this framework for building sustainability assessment should be able to provide an interface between dimensions of sustainability and phases of the building life cycle handling sustainable development challenges.

**Table 4.** Division of indicators in all the five dimensions within eight building rating tools.

| Building Tools | Sustainability Dimensions | | | | |
| --- | --- | --- | --- | --- | --- |
| | Environment | Social | Economic | Culture | Governance |
| LEED | Location and transportation Sustainable sites, water efficiency Energy and atmosphere, material and resources, indoor environment quality | Location and transportation Material and resources Regional priority | Management | | Integrative process |
| BREEAM | Health and wellbeing, energy Transport, water, material Waste, land use and ecology Pollution | Health and wellbeing Transport | Management | | |
| GREENSTAR | Management, indoor environment quality, energy, transport, water Material, land use and ecology Emissions | Indoor environment quality, transport Material, emissions | | | Management |
| CASBEE | Indoor environment, energy Resources and material, off-site environment Quality of service | Quality of service On-site environment | | | Quality of service |
| DGNB (97) | Global and local environmental impacts Resource consumption and waste Quality of technical implementation Quality of construction, site quality | Health comfort and user-friendliness, functionality, aesthetic quality Quality of technical implementation Site quality | Life cycle costing Financial performance | | |
| SEED (100) | Location and transportation Sustainable sites, water efficiency Energy and atmosphere, material and resources, indoor environment quality | Location and transportation Sustainable sites, indoor environment quality | | Sustainable sites | |
| Athena | Embodied primary energy use, global warming potential, solid waste emissions, pollutants to air, pollutants to water, natural resource use. | | | | |
| Envest 2 | Resource consumption, environmental loading. | Indoor air quality | Whole life costs | | |

**Table 5.** Proposed conceptual frameworks.

| Dimensions of Sustainability | Indicators | Assessment Stage of Building Life Cycle | | | | |
|---|---|---|---|---|---|---|
| | | Design | Construction | Operation | Renovation | Demolition |
| Environment | Sustainable sites | ✓ | | | | |
| | Land use and ecology | ✓ | | | | |
| | Material | ✓ | ✓ | ✓ | ✓ | |
| | Energy and atmosphere | ✓ | ✓ | ✓ | ✓ | |
| | Water efficiency | ✓ | ✓ | ✓ | ✓ | |
| | Emissions | ✓ | ✓ | ✓ | ✓ | |
| | Waste | ✓ | ✓ | ✓ | ✓ | |
| Economic | Life cycle costing | ✓ | ✓ | ✓ | ✓ | ✓ |
| | Professional fees | ✓ | ✓ | ✓ | ✓ | ✓ |
| | Employment | ✓ | ✓ | ✓ | ✓ | ✓ |
| | Flexibility | ✓ | ✓ | ✓ | ✓ | ✓ |
| Social | Health and wellbeing | ✓ | ✓ | ✓ | ✓ | ✓ |
| | Location and transportation | ✓ | ✓ | ✓ | ✓ | ✓ |
| | Regional priority | ✓ | ✓ | ✓ | ✓ | ✓ |
| | Aesthetic quality | ✓ | ✓ | ✓ | ✓ | |
| | Indoor air quality. | ✓ | ✓ | ✓ | ✓ | |
| | Education and awareness | | | ✓ | | |
| Culture | Artistic creation | ✓ | ✓ | | ✓ | |
| | Protection of cultural heritage | ✓ | | | | |
| Government | Management | ✓ | ✓ | ✓ | ✓ | ✓ |
| | Resilience (adaptation and mitigation) | ✓ | ✓ | ✓ | ✓ | ✓ |

This conceptual framework (Table 5) was developed through collaboration of scientific and theoretical knowledge and composed of different levels. At the first level, different sustainability dimensions including environment, economic, social, along with the addition of cultural and governmental dimensions were placed in the framework. In the second level, all indicators were placed according to each dimension of sustainability. A third level was developed to highlight the stage of the building life cycle, including design, construction, and operation till the demolition phase; excluding the predesign and recycling phase, at which an indicator needed to be assessed. As all these indicators were collected from literature, so another level (fourth level) was added in the framework in which literature references were provided against each indicator. These levels give an overall view of the sustainability dimensions related to its indicators and its importance at different stages of the building life cycle.

This conceptual framework was verified through semi-structured interviews. These interviews were undertaken involving experts who have been involved in the development of building rating tools, experienced professionals, and academics. The actual framework was developed through a collaboration of scientific and theoretical knowledge amongst different sustainability dimensions and different local stakeholders including practitioners and academia, balancing all the different dimensions of sustainability in a sequential integrative process, linking all the phases of the building life cycle. Each part was then subdivided into different parts. The detail of each section, with analysis, is described in the following section.

### 2.1.7. Interview Data Collection

The interviews were conducted at a place and time that suited the participants to make them feel at ease and encourage them to contribute and express their perspectives, feelings, and experiences in an uninhibited way. Some of the interviews were taken electronically

through online applications such as Skype. The background and their work experience are given in Table S1. This provided more flexibility and ease of giving interviews for the participants. The interviews recorded digitally lasted for approximately 45 min and were transcribed later.

Through NVivo software, data analysis conceptualized large amounts of data into smaller, more manageable pieces. Various themes/categories were made and the responses from the interview questions were grouped under the specific themes. The text was coded to retrieve answers, for example, when the researcher asked the question "Do you think that Pakistan needs to highlight and give priority to local sustainability issues and reflect them in the rating tools?" all of the answers from participants were coded as "Rating tools in Local Context of Pakistan". Each node had sub-nodes and there was a total of 25 nodes under a higher-level node named "Rating tools in Local Context of Pakistan", all of which were related to the respondent's answers about the current building rating tools implemented in the buildings construction industry of Pakistan.

In addition to this, thematic analysis was kept open to any important and significant extractable theme. According to Rose and Johnson [100], the verification procedure encompasses rich and thick descriptions of the cases, presenting negative or discrepant information that runs counter to the themes, and academic adviser's auditing. Thus, the indicators and criteria that were falling under the specific themes, cited during the interviews, were considered verified. Four thematic layers of findings were revealed in the transcripts. The four themes are enlisted as Local Context, Dimensions of Sustainability, Building Life Cycle, and Framework Development, respectively.

## 3. Interview Data Analysis Results

Although Pakistan has established its own building rating tool, SEED, the respondents confirmed that they were unsatisfied with the application of SEED relevant to Pakistan. According to respondents, SEED fully replicated LEED and many components of sustainable development that should be monitored during the assessment were missing in the guidelines of SEED. So, in a local context, it was not proving itself as the best rating tool and needed further improvements.

### 3.1. Dimensions of Sustainability Covered by the Rating Tool

Respondents were asked the question, "Do you think, three dimensions, i.e., environmental, economic, and social issues are enough to cover all the sustainability-related problem in Pakistan or we should add more dimensions to cover the sustainability issue in Pakistan more holistically?" The views of respondents were quite consistent. Most of the respondents stated that the rating tools implemented in Pakistan are not covering all the dimensions of sustainability. For instance, cultural and governmental dimensions are not considered in the rating tool. SEED rating tool only covers the environmental, social, and economic dimensions of sustainability. There is a need for coverage of additional dimensions by building rating tools.

Figure 3 highlights the responses of respondents about the dimensions that must be covered by the rating tool. It can be concluded by the views of respondents that three dimensions, i.e., environmental, social, and economical should be considered while designing the rating tools in Pakistan, however, the governance and culture dimension is also essential and should not be missed in the rating tool. Many of the respondents stated that there was a requirement to add governance and culture in the tool as shown in Figure 4. The respondents further confirmed that each dimension has its importance in a building rating tool and are linked to other dimensions (Figure 3). Respondents stated that government must take the initiative to implement the policies, to make bylaws better in the way that everyone starts using guidelines, make their buildings more livable and energy-efficient; this will also help the economy to grow and culture to preserve. The response of building stakeholders can be summed up as it is fundamental to include all the dimensions of sustainability in building rating tools that are being established for

the localized purpose. Besides, building stakeholders also suggested that the rating tools should monitor the whole building life cycle.

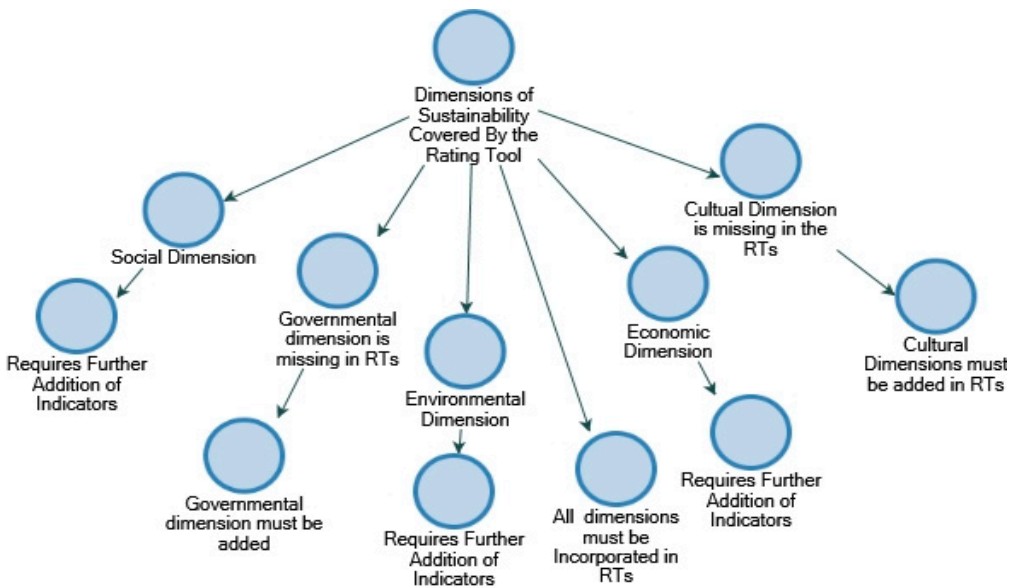

**Figure 3.** Responses of interviewees on dimensions of sustainability covered by rating tools.

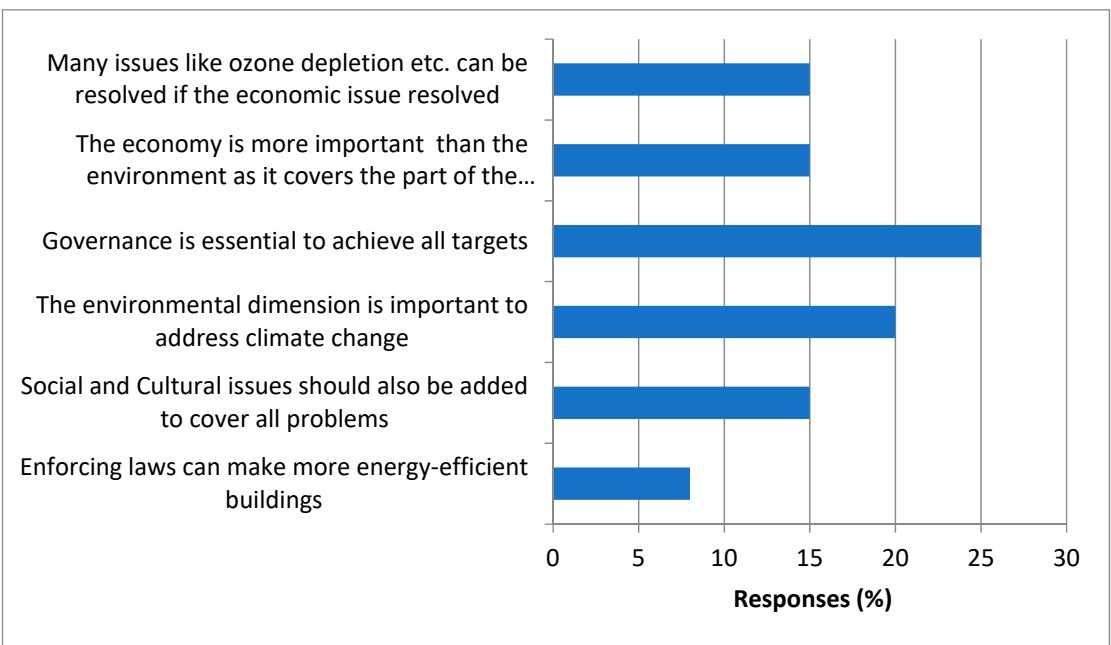

**Figure 4.** Summary of importance of all the dimensions in the rating tool to move towards sustainable development recorded in the interviews.

### 3.2. Building Life Cycle

Respondents were asked a question about life cycle thinking adoption, "Which stage of building life cycle should be assessed by the rating tool?" Respondents highlighted five stages of the building lifecycle that must be included and assessed by the rating tool including design, construction, operation, renovation, and demolition. In Pakistan people keep on living in buildings that are not deemed to be safe for occupancy. There are no quality control regulations by the government to assess the life cycles of buildings. Building stakeholders pointed out that sustainable development cannot be achieved without

monitoring the building throughout its life. Many of the respondents stated that if the five stages of the building life cycle are not considered, it is quite possible that applying sustainability rating at one stage moves the adverse effects into other stages.

Hence, all the respondents verified that it was important to monitor building throughout its life, to get productive results in terms of sustainable development. However, the respondents explained the importance of life cycle thinking adoption in more detail that could be helpful for further research, as explained in the next section.

Importance of Building Life Cycle Adoption

According to the respondents, building life cycle adoption in the rating tools is necessary to lead towards sustainable development, as it improves productivity, health, low energy bills, lesser maintenance, and renovation costs (Figure 5). The life cycle thinking adoption helps in the collection of data which can help in making positive decisions and will help in decreasing the cost in the long run as one of the respondents stated:

> *"Lifecycle assessment will help in data collection for decision making which will help to make a positive decision and save our local materials and cost less in the long run with less electricity usage."*

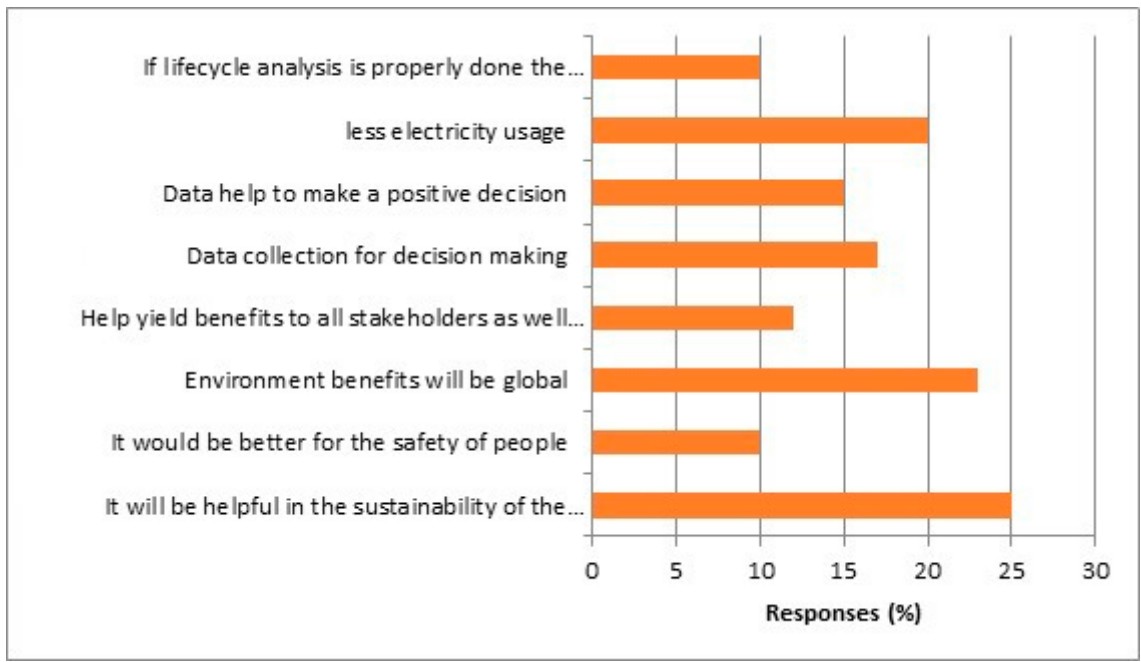

**Figure 5.** Importance of lifecycle adoption in the rating tool responses summarized.

According to respondents, exposure to life cycle thinking in a building will help in decision making. Everyone in the entire life cycle chain of a building, from cradle to grave, has a duty and a position to play, taking into consideration all the related cultural, environmental, and social impacts. The analysis showed that the benefits will be enormous if life cycle thinking is adopted in the assessment process along with addressing the energy scarcity issue. According to respondents, in the next few decades, the benefits related to the environment, especially energy-related, will be global. Studies have indicated that with the application of sustainability practices an increase in economic activities and associated jobs will increase [84].

Respondents also highlighted other benefits which include better lifestyle, water conservation, and less consumption of electricity. The important stages of the life cycle for the assessment process and the barriers to its adoption as per the respondents' responses are given in Figures 6 and 7. Transcript analysis shows that five stages of the building life

cycle, including design, construction, operation, renovation, and demolition, are important to be assessed by rating tools. The major barrier in life cycle thinking adoption pointed out by respondents includes governmental and economic issues. The doubts of different stakeholders and contractors need to be addressed. Respondents argue that it is essential to check the compliance to green standards or rating tool that the building is designed to follow.

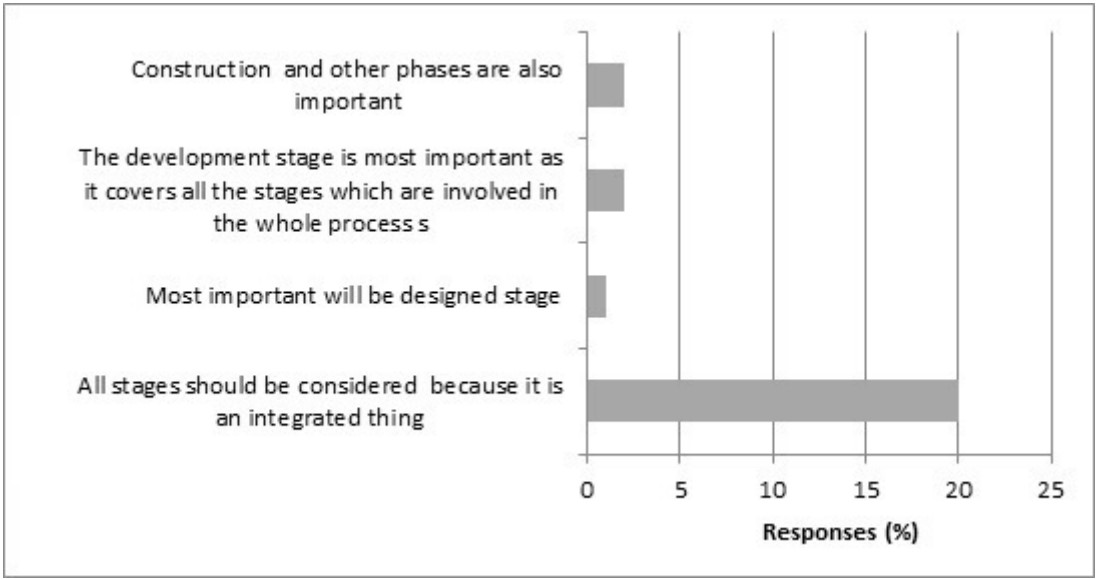

**Figure 6.** Important stage of the life cycle for the assessment process in the rating tool recorded in the interviews.

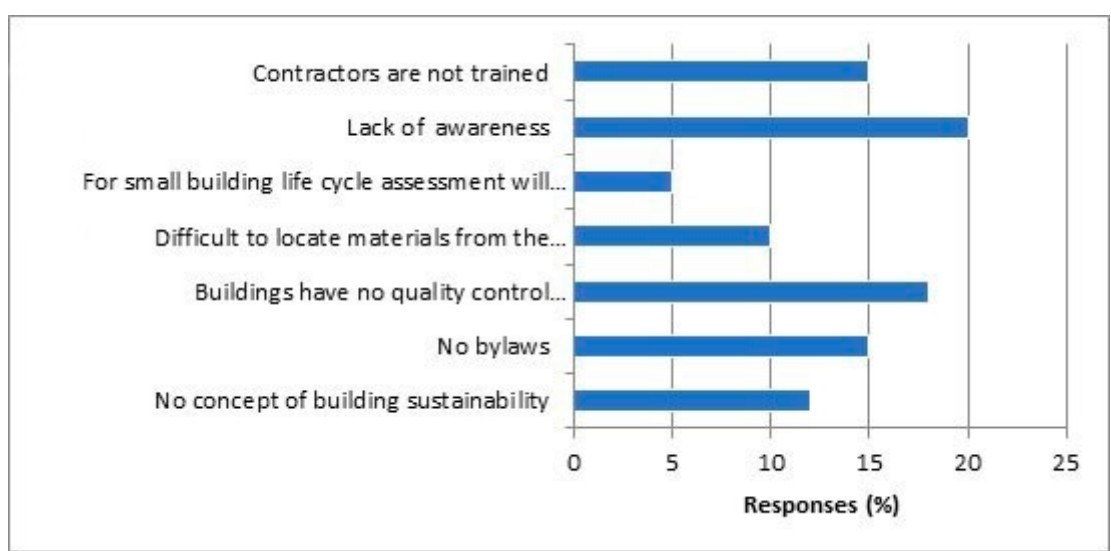

**Figure 7.** Barriers in life cycle adoption.

An overall analysis of interview transcripts showed that the rating tools were not localized, sustainability dimensions were not covered by rating tools, and buildings were not being monitored at all stages. In this regard, respondents suggested some recommendations in the form of a framework, that if developed could help stakeholders and governmental organizations to establish localized rating tools to lead towards sustainable development. This framework is described below.

*3.3. Framework Validation*

The conceptual framework developed through literature review was validated by interviewees. Respondents were provided the developed framework and questions were asked regarding different indicators. Respondents explained the importance of each indicator with a suggestion for adding, refining existing, and omitting some overlapping indicators. The validated framework is explained below in detail.

Validation of Indicators

Respondents were asked questions about the inclusion of different indicators, proposed in the conceptual framework, to be a part of the rating tools. The indicators included land use and ecology, sustainable sites, material and resources, indoor air quality, acoustic, employment, life cycle costing, health and wellbeing, protection of cultural sites, management, resilience, etc. [101,102]. Varying responses were received for these indicators against five stages of the building life cycle, i.e., design, construction, operation, renovation, and demolition.

According to the respondents, the first indicator that should be considered in the design stage was found as "Land" as the response given below:

> *"Whenever we design a thing/building, we consider the land, keep the land in our mind and try to design the building in such a way that we minimize land use and use it functionally. In construction, we try to reduce the impact on land, whenever we are constructing, we use heavy machinery which has a direct impact on the land. We use the water and the topsoil washes away if we do not consider good techniques in the design phase. And demolition phase, whenever we demolish a building if we don't clear the debris it goes to the land."*

Respondents validated that the second indicator considered for including in the framework should be "Material", as most of the respondents suggested this as an important indicator. As locally produced material is the social aspect benefitting the society with its increasing demand and utilization. Other indicators that were verified against the environmental dimension were energy and atmosphere, water conservation, emissions, and waste management. According to respondents, the inclusion of water, energy, and atmosphere can aid in improving the sustainability of the building and the wellbeing of people working in it. Constructing a building with water-efficient equipment, kitchen gardens, and reduced waste will result in a cleaner environment. A clean atmosphere with clean emission practices is also important in all phases of the building life cycle.

In the economic dimension "investment cost" was added as the first indicator that was a refined form of "life cycle cost". According to respondents, to make the buildings economically sustainable it is necessary to increase the investment costs that cover all the life cycle costs necessary to adopt sustainable practices. The other main indicator validated against the economic dimension of sustainability by the respondents was employment. All the stages of construction will need manpower and it is up to the management how many staff will be employed for the project.

Respondents denied the "professional fees" as according to them employment also covers this indicator. The last indicator validated by the respondents in the economic dimension was the flexibility in the design of a building. Overall, regarding the environment and economic dimensions, the respondents' view was that a holistic assessment will assure these burdens are not transferable to other stages or stakeholder but are rather eliminated from building a life.

According to respondents', social indicators should also be included in the five stages of the building life cycle. As stated by one of the respondents:

> *"Social equality, equity, social balance, for instance, the satisfaction of people with the building, health and wellbeing, location and transportation for employee's movement, aesthetic quality and indoor air quality are also important indicators. All of these must be considered in the framework for the people living inside the building and people who are*

*outside the building, so they are not affected by the building. For instance, if a building overshadows another structure that is nearby you see you are affecting people, you are affecting society so that also counts as a major indicator of the framework."*

Respondents omitted the indicator "Regional priority" as according to their views the proposed conceptual framework is already based on the local context of Pakistan so there is no need to include it in the framework as an indicator. Another social indicator validated by the respondents was "education and awareness". There is a need to create awareness "*about energy efficiency, sustainability, energy usage, water conservation, waste management at all stages of the building life cycle. Social marketing and social advertisement and social awareness are very important to lead towards sustainable development.*"

Respondents added "Safety and Security" as another important social indicator that was not proposed in the conceptual framework. One of the respondents commented on this as:

*"I think in Pakistan the security of the buildings and usage of sensors can be very useful for security purposes. For example, whenever there is someone tries to come into the buildings these sensors will turn on, and security will be able to easily take an action, in this way these sensors will help the security of the buildings."*

Respondents validated "heritage conservation" as an important indicator of the framework and omitted artistic creation as an indicator. Many of the respondents pointed out that heritage has completely destructed in our country, it is important to include it in the framework. Respondents added another cultural indicator in the framework named "cultural identity". Cultural identity must be considered in the design phase if the construction aims to reflect it, as it must be designed accordingly to make it culturally rich and help to grow economically.

Respondents added two more indicators including, "policies and regulations" and enforcement". As the respondents confirmed that the "*Governments needs to develop novel regulatory tools for buildings sustainability and resilience and industries need to implement such governance and regulatory tools*".

Respondents commented that for a holistic approach there must be regulations and policies in the framework in all stages of the building life cycle, to advance the management of construction, operation, renovation, and demolition decisions. It will reduce the environmental, social, and economic impacts on the whole industry. "The government presently lacks the infrastructure for the management of building life cycle".

According to respondents, the government does not implement any rules and regulations, which are a necessary component for sustainable development. Strict actions by the government can help in enforcing the policies in achieving sustainable development and improving management of different stages of the building life cycle. The government needs to encourage the usage of advanced materials that improve resilience in building infrastructure. Based on the responses, the proposed conceptual framework is validated (Figure 8), with the addition, omission, and edition of some of the indicators. Assessment of each indicator through the different phases of the building lifecycle will help in better uptake and in achieving sustainability.

According to respondents, this validated framework is a holistic approach that will enormously benefit a move towards sustainable development, as shown in Figure 8.

With the help of this framework, better-informed decisions can be made that have long-running impacts, not only on the environment but also on the health of the people. The benefits can be wider implications on the whole socioeconomic context. Table S2 outlines the indicators that have been added, refined, or omitted after interviews with the stakeholders.

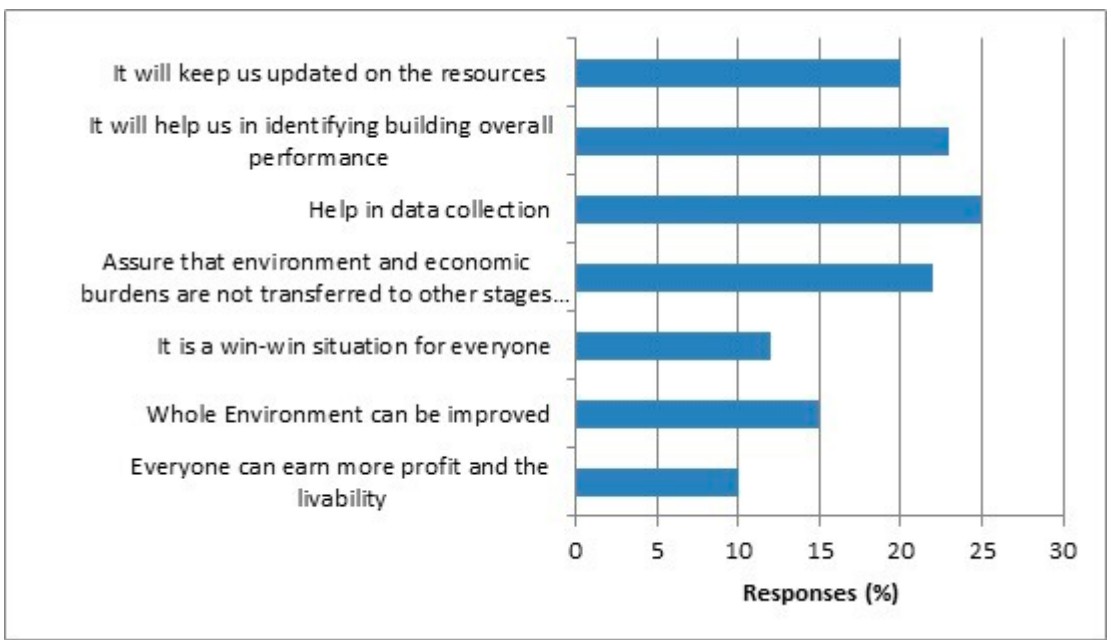

**Figure 8.** Benefits of a holistic framework as suggested by respondents.

## 4. Discussion

The views of all the respondents were consistent in the local context of the rating tool as all the respondents have extensive experience and knowledge on green building rating tools development or sustainable development. Overall, it can be stated that, according to respondents, rating tools implemented in Pakistan, such as SEED, are not working appropriately in the context of Pakistan, which is supported by the study of Ullah, Noor [15]. SEED fails to address the local context and challenges of Pakistan. Regional priority should be allocated more credit so that the rating tool can set its challenges and targets. Otherwise, the assessment outcomes may mislead the sustainability targets. The SEED rating system emphasizes more on the environmental aspect of sustainability, without giving due consideration to the economic aspect which is important for Pakistan. In a developing country like Pakistan, economic and social aspects are significant. The government was quoted as a major hindrance to the development and implementation of green building rating tools in Pakistan. Pakistan's government consists of fragmented political parties, due to the higher corruption rate there is no focus on the development of the country. There is a lack of legislation and regulations by the government. It is necessary to upgrade the governmental system to lead the country towards sustainable development. There is a need for focused research on building rating tools to improve the system. In the literature, it is reported that the government plays an effective and undeniable role in the development of green building rating tools [19,101,103,104]. However, respondents suggested that improvements in the governing structures of the country can improve the patterns of rating tools to move towards sustainable development.

The views of respondents for sustainability dimensions can be summed up as follows; in addition to environmental, economic, and social dimensions, there must be an addition of dimensions of sustainability such as culture and government in the building rating tool to address the unique cultural identity and governance issues in Pakistan. A study conducted by Al-Jebouri, Saleh [103] stated that five fundamental dimensions of sustainable development (environment, social, culture, governance, and economy) can drive sustainability in the building sector, reduce duplication of efforts, and increase the opportunities for innovative practices. Many researchers have recommended the adoption of these additional dimensions into the green building rating tools. For example, Fatourehchi and Zarghami [28] defined four distinctive parameters for assessing alternative scenarios,

including economic, environmental, technological, and social parameters. Hence, the adoption of this comprehensive perspective in the rating tool is significant to assess the sustainability of any building in Pakistan.

The respondents also emphasized the life cycle thinking adoption in all stages. Many of the respondents stated that if all stages of the building life cycle are not monitored, it is quite possible that issues produced at one stage move the adverse effects to other stages. This is in line with the findings of by Mazzi [83] and Peris Mora [84] indicating the use of life cycle assessment methodology in the building design stage which will facilitate policymakers to evaluate the life cycle impacts of building products, components, and materials. This helps to choose the materials and products in a way that the building life cycle impacts are reduced. Ikhlayel [82] claimed that the development of life cycle-based tools is on a continuous rise in the building sector. These tools are used in the building sector in two ways; one is to use these tools to access the building product throughout their life cycle and the other way is to access the building. However, in Pakistan, there are many hindrances in the adoption of this concept in the rating tool, according to the survey conducted. The major issues pointed out by respondents for life cycle adoption includes governmental and economic barriers. A study by Ding, Fan [101] supports this view as it is the responsibility of the government to provide support for the adoption of green practices by providing incentives to the public. Hence, these statements suggest that the government should implement strict rules and regulations and build new strategies for green building rating tools.

Respondents validates the need for the conceptual framework and showed why each validated indicator is important to be included in it. According to them, it will be beneficial to assess each indicator for different phases of the building lifecycle to achieve more sustainable infrastructure. In general, it was stated that the framework can prove itself as an instrument to rate the building in every step/phase from design to demolition, it would be good because if this framework will be followed it can help future researchers to establish the more localized rating tools, cover all the dimensions of sustainability, and adopt life cycle thinking approach. Furthermore, the inclusion of key indicators in the framework was suggested which best suits the regional characteristics for any sustainability analysis and encourages sustainable development at the local level.

## 5. Conclusions

This study finds that a localized green building rating tool is important for the sustainable development of the building sector of Pakistan. Application of all the dimensions of sustainability, i.e., environment, social, economic, culture, government, with due weightage as per local requirements, will potentially enhance sustainable development in the region and reduce the negative impacts. Localized rating tools will potentially assure that the buildings are more sustainable and adaptive in the local environment. It will enable policymakers and management authorities to provide long term effective solutions for highly complex systemic problems faced by Pakistan. The rating tools are not undergoing any revisions and updates because of a lack of governmental support and bylaws implementation. Particularly, it is emphasized that to move towards sustainable development the government needs to take steps to build new strategies for the improvement of rating tools.

A novel framework based on the quintuple bottom line model in conjunction with building lifecycle has been developed for the Pakistan building industry. This proposed framework has been developed and validated as an outcome of this research, by integrating criteria from the analysis of transcripts (Table 5). Overall, this framework, consisting of five dimensions of sustainability along with the five stages of the building life cycle has been proposed for the development of localized rating tools. Therefore, this comprehensive framework should be adopted to meet the regional characteristics and sustainability analysis for the local building industry. In this framework, almost twenty key indicators such as land, material, air quality, acoustic, health and safety, employment, policies, regulations,



enforcement, heritage, cultural identity, etc., have been included. These indicators are differently placed against each stage of the building lifecycle as shown in Table 6. Adoption of this framework can help multi-stakeholders to:

- Develop new localized approaches.
- Develop new sustainability goals and strategies for the country according to its issues, culture, and practices.
- Learn from each other's ideas and work to develop new strategies for sustainable development of buildings.
- Develop up-to-date frameworks that are based on technical and scientific research.

**Table 6.** Proposed conceptual framework.

| Dimensions of Sustainability | Indicators | Assessment Stage of Building Life Cycle | | | | | References |
|---|---|---|---|---|---|---|---|
| | | Design | Construction | Operation | Renovation | Demolition | |
| Environment | Land Use | ✓ | | | | | [44] |
| | Material & Resources | ✓ | ✓ | ✓ | ✓ | | [45] |
| | Energy & Atmosphere | ✓ | ✓ | ✓ | ✓ | | [26] |
| | Water efficiency | ✓ | ✓ | ✓ | ✓ | | [9] |
| | Emissions | ✓ | ✓ | ✓ | ✓ | ✓ | [38] |
| | Waste management | ✓ | ✓ | ✓ | ✓ | ✓ | [74] |
| Economic | Investment cost | ✓ | ✓ | ✓ | ✓ | | [12,15] |
| | Employment | ✓ | ✓ | ✓ | ✓ | ✓ | [26] |
| | Flexibility | ✓ | ✓ | ✓ | ✓ | ✓ | [3] |
| Social | Health & Wellbeing | ✓ | ✓ | ✓ | ✓ | ✓ | [42] |
| | Location & Transportation | ✓ | ✓ | ✓ | ✓ | ✓ | [65] |
| | Acoustics | ✓ | ✓ | ✓ | ✓ | ✓ | [76] |
| | Safety and security | ✓ | ✓ | ✓ | ✓ | ✓ | [67] |
| | Indoor Air Quality. | ✓ | ✓ | ✓ | ✓ | | [72] |
| | Education and awareness | ✓ | ✓ | ✓ | ✓ | ✓ | [79] |
| Culture | Cultural heritage conservation | ✓ | ✓ | | ✓ | | [38,55] |
| | Cultural identity | ✓ | | | | | [9,59] |
| Government | Management | ✓ | ✓ | ✓ | ✓ | ✓ | [20] |
| | Resilience | ✓ | ✓ | ✓ | ✓ | ✓ | [3] |
| | Policies & Regulations | ✓ | ✓ | ✓ | ✓ | ✓ | [12,54] |
| | Enforcement | ✓ | ✓ | ✓ | ✓ | ✓ | [54] |

This framework is a holistic approach that draws the attention of future researchers towards the importance of adapting the needs and considerations of local aspects in any emerging rating tool. In future research and developments, the framework can be further advanced by assigning the appropriate weightages to the identified sustainability assessment indicators. Furthermore, green building rating tools could be assessed by applying them to a case study in Pakistan. Moreover, a future study can develop local codes and standards, based on the indicators provided in this framework, through stakeholder's participation and technical knowledge. This framework can also facilitate estimating the workability of the sustainability assessment framework through case studies in Pakistan.

**Supplementary Materials:** The following are available online at https://www.mdpi.com/article/10.3390/buildings11050202/s1, Table S1: Background and Experience of Respondents and Table S2: Indicators refined after stakeholder consultation.

**Author Contributions:** Conceptualization, M.A.K., C.C.W.; Methodology, M.A.K., C.C.W., C.L.L.; Analysis, M.A.K., C.C.W.; Discussion, M.A.K., C.C.W., C.L.L.; Writing and revision, M.A.K., C.C.W., C.L.L. All authors have read and agreed to the published version of the manuscript.

**Funding:** This research receives no external funding.

**Institutional Review Board Statement:** Not applicable.

**Informed Consent Statement:** Not applicable.

**Data Availability Statement:** Data can be made available on request.

**Conflicts of Interest:** The authors declare no conflict of interest.

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
