# Peer review of "A Framework for Developing Green Building Rating Tools Based on Pakistan’s Local Context"

_buildings, doi:10.3390/buildings11050202_

Round 1

Reviewer 1 Report

The research presented in this paper concerns the development of a holistic framework of building rating tool incorporating cultural and governmental dimensions based on Pakistan’s local context. This is the main originality of this research.  

This is the main originality of this research justifying a possible publication.

But this paper cannot be published in its actual form. It is too long   with too many tables and figures. Use annexes or put some data, figures and tables in supplementary material. There are also some major remarks:

L78 Why nothing about methods for interview and results analysis in this paragraph?

L84: Table 1 to delete because unuseful

L86  Title of Table 2  to put before the table  and I am not sure that the Table2 is pertinent.

Give more explanations about your literature review analysis:

L146 Re-organize the Table. In the first column, group identical items   and in the second column indicate the corresponding references such as :

Social, Economic, Environment and Governance   Woermann and Engelbrecht [34] ; Alibašić [42]…

In your analysis of the literature, highlight the historical evolution of the perception and measurement of sustainable development. You can clarify these terms: pillars, domains or dimensions to avoid confusion.

L192 3.4:  The proposal to integrate five pillars should be more clearly presented and with more importance in the paper due to its originality. Contrary to what the authors say, this evidence is not so clear in the literature. What is the difference, for example, with the United Nations approach integrating economic, ecological, political and cultural dimensions?

L316 and Table 7 Governmental dimensions is unclear for me. What's difference with governance?  Define more clearly the items : management and resilience, in this pillar.

L340 Table 8 in annex or as supplementary material

Give more information about the choice of the questions and the respondents and also method used. There are only 25 respondents?  What reliability and representativeness can we give to the survey in this case?

Reviewer 2 Report

In the contemporary progress of design principles in the urban and architectural dimension, the way of control is very important. Certifications are one of the ways of such control and an indicator of correctness both at the level of planning and implementation of structures around and strictly building, therefore their role is very important. They provide information that inspires confidence in the implemented object or space, giving rise to the opinion that we are dealing with sustainable development that meets pro-environmental expectations while respecting the principles of ergonomics and the art of construction. The article compares criteria used in the preparation of such documentation prepared in other countries (including GREENSTAR, CASBEE, DBNG) or implemented locally (SEED) the most popular systems (LEED, BREEM). Using the example of regulations adopted in Pakistan, he tries to take a critical look at this documentation, making use of various expert opinions. On this basis, Authors builds a framework for an even more thorough inspection system, including above all social and cultural aspects, marginalized in existing certification systems. A major contribution here is the emphasis on the very humanistic factor of "Life Cycle Thinking". It is important, because it refers to the very human dimension of existence and habitation, which, as the authors show, has a huge and obvious impact on the quality of the assessed objects and urban structures. Referring to the above opinion, I consider the article to be very good, consistent with the profile of the magazine Buildings (although perhaps even more fitting for the journal Sustainability, within the publishing house MDPI). Such texts are needed and should appear in the scientific space, which does not mean, however, that it did not avoid minor errors:

The introduction clearly shows the research background relevant to the contents of the manuscript. General differences in approaches to the problem of certification in the world are presented here. Research theses and objectives of the presented research are clearly described. The structure of the developed text has been described in a general but sufficient way.

Chapter 2 presents a concrete and factual breakdown of the structure of the manuscript, highlighting the presented content with well-prepared tables that synthetically show the logical arrangement of the text's construction. The order of tables in the text and their captions raises more editorial than substantive doubts (the numbering of Tables is also incorrect - there are 1, 2 and then 4). Note concerns L86 - 93 - the location of the signature should be under Table 2.

Chapter 3 presents a literature review highlighting aspects of the culture and social dimension of certification. The introduction showing different certification approaches to understanding sustainability, tabulated, is acceptable. The table itself is very interesting - clearly synthesizing the literature review, a minor editorial error is visible (L 136 - 147 - numbering error in the text and in the table caption). A very important and innovative element is the description here of the idea of "Life Cycle Thinking" and its benefits in the real use of the assessed buildings. The differentiation of aspects taken into account in the analyzed types of certification shown in Table 6 is interesting. The interpretation of the presented library search in Table 7 as a framework list, being a starting point for further research, is also impressive. Doubts are again raised on the editorial side where chapter (sub-chapter ?) 3 (L 333) does not agree with the rest of the numbering. This needs to be checked. On the other hand, the information contained herein relating to surveys carried out among various social groups connected with the creation and construction, as well as use of buildings, is very interesting. The results shown, which draw attention to the social value of cultural and psychological elements as a direct factor in the economic success of a project, are unusual and important. It is a little confusing to include a content here referring to the results. It would be advisable to rethink this part of the text structure and compare it with the contents of Chapter 4. Figure 4 is a little illegible (L 357) technical editing would be advisable. The current poor resolution makes it a bit unreadable.

Chapter 4 focuses on research findings taking into account the local context of Pakistan. In general, the contents presented follow the structure presented in Chapter 2 and are unlikely to raise questions. Of particular interest is chapter 4.4.1, which describes the validation of indicators. It is very well summarized in Table 9 - which is a very clear summary of the presented results (improvements are due to its technical embedding in the text) and shows how certification schemes should evolve. Figure 5 (L 382) is a bit unclear - it does not quite match the description below. It would be worth revisiting this Figure and perhaps rewording it.

The discussion presented in Chapter 5 is clear and relates well to the literature review presented in Chapter 2 and the introduction.

The conclusions presented in Chapter 6 are clear and concrete. They are very well complemented by Table 4 which is an excellent summary of the research described in the text. The content of this chapter is a good starting point for further work on certification of buildings and urban structures.

Reviewer 3 Report

This is a generally well-written paper on an important topic that is of much relevance to the A/E/C industry. The authors do a good job of identifying the problem they would like to address (the need to contextualize green building rating system based on the specific and local conditions of a region), which, indeed, is important. However, there are numerous issues that the authors need to address:

1) The most important issue is that the paper sets out to develop a new framework to address the local conditions of Pakistan; and yet the developed framework just seems to be based on a number of existing green building rating tools; without having anything specific to Pakistan. In other words, it is not clear how/why the developed framework, with the items shown in Table 10 would be specific to Pakistan, if they were simply adopted from other rating tools? Similarly how is this framework different from the SEED system that is already developed for Pakistan, which the authors state to be as too general (“The developed rating tool SEED is based on western standard and is not able to address the local environmental needs and sustainability concerns”) in their point of departure. Without having a solid discussion on the uniqueness of the framework they develop as well as its ability to address local conditions, the authors cannot justify that they have achieved their purpose of developing a new green building rating tool that considers the needs specific conditions of Pakistan. I am not suggesting that the authors should have started from scratch in developing their framework; making use of the existing ones is a good idea; but how their end product is different from the existing ones and addresses the specific conditions of Pakistan is unclear.

2) Along the lines of my comment above, why develop a new framework when one can adopt and adapt an existing framework to meet the local conditions of a region?  How about looking at the existing ones and changing those as opposed to developing a brand new framework effort? And even though the authors claim to be developing a brand new framework; they seem to be just adopting and adapting the existing ones.

3) The authors state that “Based on the local context of Pakistan each indicator and category are ranked according to their importance.” I don’t see any ranking in this paper?

4) The statement “The developed rating tool SEED is based on western standard and is not able to address the local environmental needs and sustainability concerns” is an opinion unless supported by references. And since this is the point of departure of this research, this statement needs to be supported.

5) The interview question/answers about the addition of two more dimensions to the assessment of sustainability are confusing. The authors suggest that they asked an open-ended question as to whether additional items other than the triple bottom line should be considered and that the respondents suggested the addition of governance and culture. It seems odd to me that the respondents suggested the addition of the exact two items that the authors had already identified in their draft framework (that they wanted to validate through the interviews) and makes me wonder if the respondents were steered towards providing those two item either by giving them options or suggesting those during the interviews. The authors should explain this.

6) The paper has a good literature review; though it is too long for a paper of this nature. This is not a review paper. The authors should only include the most relevant portions. To me, the idea of adding the 4th and 5th bottom line to measure sustainability (governance and culture) deserves the most attention. The rest of the literature review is widely available and should only be cited as necessary.

7) Table 8 is unnecessary. The authors should only summarize the qualifications of the interviewees.

8) Figures 2,3,4,5 are not necessary and can easily be discussed in the text.

9) The acronym TBL should be spelled out the first time it is used.

10) Future research can be included as a paragraph under the Conclusions section, as opposed to being presented as a sub-section (6.1).

Reviewer 4 Report

Introduction
The paper is well introduced concerning the environmental impacts generated by the construction industry worldwide and then tackles the different building rating tools present in all countries to provide green certification standards. In Pakistan, a building certification tool has been developed, known as SEED, based on Western countries' environmental, economic, social, and cultural parameters. The authors argue that this tool is inadequate for Pakistan and explain how they intend to approach the study (literature review and data analysis) to provide suitable building rating tools to gauge the green building criteria.
The methodology of the green building rating tools is based on considering many aspects that should be calibrated in individual countries. In the case study, for example, there is no merge between GBRT and life cycle assessment. Therefore, there are gaps in the evaluation tool.
Line 56: I don’t find the refernce in the bibliography. Please check.
Lines 71-72: this concept is already mentioned above (lines 65-70), it is redundant.
Materials and Methods
Line 84: Shift the caption of Table 2 above the table.

Literature Review
Literature Review is one of the survey methods, along with stakeholder interviews. For this reason, perhaps it could become a subparagraph (2.1) of the Materials and Methods paragraph. In my opinion, an extensive review of literature has been made, which, however, should be slightly summarized. The implementation of the indicators-pillars of sustainable development to have a holistic, transversal approach focused not only on the environment but also on governance and cultures is interesting. The quintuple bottom line approach and the evaluation of life cycle thinking referring to the five main phases of a building, from design to its demolition, lead to Pakistan's conceptual definition of sustainability.
Line 136: TBL if you use the acronym, you have to explain it.
Line 155: please delete that

3.6 Key indicators
The sustainability indicators were selected from the literature and were evaluated within each phase of the building's life cycle.
Interview data collection
The numbering of paragraphs and subparagraphs is to be checked. Please check them.
Is table 8 necessary? I am not sure because the profiles of the participants have been explained above.

4. Interview data analysis results
Graphs 7, 8, 9, 10 need to be improved: please remove the graph title from the left side. The captions should help illustrate the graph.
Line 442: The citation is not made according to the guidelines of the Journal. Please correct.
This part can be improved with a more accurate scientific language.
Line 483-485: regarding the minimization of land consumption, you should mention this report in the references: https://sustainabledevelopment.un.org/content/documents/5987our-common-future.pdf
The Bruntland report offers a perfect definition of sustainable development; please check pages 42-43.

Discussion
In this section, the main topics analyzed in the previous sections are discussed homogeneously. It is considered appropriate to implement government strategies aimed at sustainable development; it is correct to include culture and governance among the pillars of sustainability and to evaluate the indicators in all phases of the life cycle of building.

Conclusion
The section discussion concepts are repeated, and table 10 does not present different concepts than tables 7 and 9. It would be interesting to foresee future research developments by applying the green building rating tools to a case study in Pakistan.
In conclusion, I suggest summarizing some sections with redundant concepts, submitting the manuscript to English editing revision, and following the authors' guidelines for graphics, tables, and references.

Round 2

Reviewer 1 Report

Acceptable under this present form

Author Response

Thank you for the recommendation.

Reviewer 2 Report

In the revised version of the manuscript, the Authors consider the most of the proposed corrections and additions. Minor corrections in the conclusions are still possible in order to improve the clarity of the main purpose of the paper. The manuscript can also be accepted in its present form. Good work!

Author Response

Thank you very much. We have further improved the conclusion section.

Reviewer 3 Report

The authors have addressed my comments. As two final minor comments:

1) It seems like there is inconsistency in the use of tense in the paper. The authors use future tense when talking about the methodology of a study they already concluded. Example: “This conceptual framework will be verified through Semi-structured interviews. These interviews will be undertaken involving experts who have been involved in the development…” The authors should perform a thorough check of the paper to address this issue.

2) The authors have chosen to add supplementary files (Table SI-1 and SI-2) to their submission; yet the paper does not provide references to those files.

Reviewer 4 Report

I have read the new version of the manuscript. I noticed that the authors accepted the suggested revisions. Therefore I have no other suggestions to provide.

Author Response

Thank you for the recommendation.